# An updated seabed bathymetry beneath Larsen C Ice Shelf, Antarctic Peninsula

Alex Brisbourne[1], Bernd Kulessa[2], Thomas Hudson[1], Lianne Harrison[1], Paul Holland[1], Adrian Luckman[2], Suzanne Bevan[2], David Ashmore[3], Bryn Hubbard[4], Emma Pearce[5], James White[6], Adam Booth[5], Keith Nicholls[1] and Andrew Smith[1]

[1]British Antarctic Survey, Natural Environment Research Council, Madingley Road, Cambridge, CB3 0ET, UK.
[2]Glaciology Group, College of Science, Swansea University, Singleton Park, Swansea SA2 8PP, UK
[3]School of Environmental Sciences, University of Liverpool, Liverpool, L69 7ZT, UK
[4]Centre for Glaciology, Department of Geography and Earth Sciences, Aberystwyth University, Aberystwyth, SY23 3DB, UK
[5]Institute of Applied Geoscience, School of Earth and Environment, University of Leeds, Leeds, LS2 9JT, UK
[6]British Geological Survey, Keyworth, Nottingham, NG12 5GG, UK

*Correspondence to*: Alex Brisbourne (aleisb@bas.ac.uk)

**Abstract.** In recent decades, rapid ice-shelf disintegration along the Antarctic Peninsula has had a global impact through enhancing outlet-glacier flow, and hence sea-level rise, and the freshening of Antarctic Bottom Water. Ice-shelf thinning due to basal melting results from the circulation of relatively warm water in the underlying ocean cavity. However, the effect of sub-shelf circulation on future ice-shelf stability cannot be predicted accurately with computer simulations if the geometry of the ice-shelf cavity is unknown. To address this deficit for Larsen C Ice Shelf, west Antarctica, we integrate new water-column thickness measurements from recent seismic campaigns with existing observations. We present these new data here along with an updated bathymetry grid of the ocean cavity. Key findings include relatively deep seabed to the south-east of the Kenyon Peninsula, along the grounding line and around the key ice-shelf pinning-point of Bawden Ice Rise. In addition, we can confirm that the cavity's southern trough stretches from Mobiloil Inlet to the open ocean. These areas of deep seabed will influence ocean circulation and tidal mixing, and will therefore affect the basal-melt distribution. These results will help constrain models of ice-shelf cavity circulation with the aim of improving our understanding of sub-shelf processes and their potential influence on ice-shelf stability.

30 The data sets comprise all the new point measurements of seabed depth. We present the new depth measurements here as well as a compilation of previously published measurements. To demonstrate the improvements to the sub-shelf bathymetry map which these new data provide we include a gridded data product in the supplementary material of this manuscript, derived using the additional measurements of both offshore seabed depth and the thickness of grounded ice. The underlying seismic data sets which were used to determine bed depth and ice thickness are available at https://doi.org/10.5285/315740B1-A7B9-

4CF0-9521-86F046E33E9A (Brisbourne et al., 2019), https://doi.org/10.5285/5D63777D-B375-4791-918F-9A5527093298 (Booth, 2019), https://doi.org/10.5285/FFF8AFEE-4978-495E-9210-120872983A8D (Kulessa and Bevan, 2019) and https://doi.org/10.5285/147BAF64-B9AF-4A97-8091-26AEC0D3C0BB (Booth et al., 2019).

## 1 Introduction

The loss of Antarctic ice shelves is of global significance for two reasons. First, ice shelves provide a buttressing force – controlled by the geometry and stress regime of the ice shelf - to the glaciers or ice streams that feed them. Although loss of the floating ice shelf makes only a small direct contribution to sea-level rise, the removal of buttressing results in acceleration of the tributary glaciers, enhancing their current contribution to sea-level rise (Rignot et al., 2004; Scambos et al., 2004; Rott et al., 2002; Fürst et al., 2016). Secondly, basal melting of ice shelves produces cold and low-salinity water that influences Antarctic Bottom Water (AABW) formation, which in turn affects the properties of the global oceans (Jacobs, 2004).

Over recent decades, there has been a southwards progression of ice-shelf loss along the eastern Antarctic Peninsula. The disintegration of the Larsen A Ice Shelf in 1995, and the Larsen B in 2002, resulted in a step increase in flow of the grounded glaciers that formerly fed these ice shelves (e.g., Khazendar et al., 2015). This increase in glacier flow resulted in accelerated sea-level rise and increased freshening of dense AABW (Jullion et al., 2013). In a number of cases, ice-shelf retreat has been attributed to atmospheric warming (Vaughan and Doake, 1996; Rott et al., 1998; Skvarca et al., 1999). With the Antarctic Peninsula exhibiting one of Earth's highest rates of atmospheric warming during the late twentieth century (Vaughan et al., 2003), the long-term viability of the Larsen C Ice Shelf (LCIS) is in question. However, Holland et al. (2015) demonstrated that the thinning of LCIS over the last decade is a result of both atmospheric and oceanic influence in almost equal measure. For the remaining ice shelves on the Antarctic Peninsula, the relative contribution to their future stability by basal melt from incursions of relatively warm ocean water, and increased surface melting by a warmer atmosphere, is still unknown.

To improve projections of the effects of basal melt on ice shelves, knowledge of the geometry of the ocean cavity beneath is vital (Mueller et al., 2012; Jenkins et al., 2010; Grosfeld et al., 1997; Goldberg et al., 2019; Pattyn et al., 2017). Models of sub-shelf circulation are critically dependent on cavity geometry, particularly in regions where the influence of strong tides is topographically constrained (e.g., Mueller et al., 2012). Ongoing efforts to model ocean processes beneath LCIS suffer from inadequate knowledge of cavity geometry because seabed depth is poorly sampled (Brisbourne et al., 2014). Improving knowledge of cavity geometry is crucial for LCIS because the sparse existing data suggest the presence of large-scale seabed features capable of guiding ocean currents and inducing significant tidal mixing. It is impossible for computer simulations to predict accurately the future influence of the ocean on LCIS without knowledge of the geometry of such features.

Although labour intensive, seismic methods remain the most reliable method for determining sub-shelf cavity geometry. Airborne and ground-based radar are used extensively to map ice thickness but cannot penetrate the sub-shelf cavity. Autonomous underwater vehicles (AUV) provide another direct measurement of sub-shelf bathymetry but with limited coverage at present (e.g., Jenkins et al., 2010). Inversion for water column thickness using airborne gravity measurements is sensitive to assumptions about local density variations such as sediment infill and may lead to inaccurate results (Brisbourne et al., 2014). Recent studies using gravity inversion combine data from multiple methods to address these assumptions (e.g., Muto et al., 2016).

## 2 Location and previous work

LCIS, the largest ice shelf on the Antarctic Peninsula at around 44 000 km$^2$ (Cook and Vaughan, 2010), lies just south of the recently collapsed Larsen A and B ice shelves (Fig. 1). The geometry of LCIS's sub-shelf cavity has previously been measured in detail at specific locations only (Brisbourne et al., 2014): this campaign was designed to target locations where an existing inversion of gravity measurements indicated areas of significant control over sub-shelf circulation (Cochran and Bell, 2012). However, uncertainties associated with gravity inversions for bathymetry result in large areas of unknown geometry, specifically beneath LCIS (i) away from the western grounding line, (ii) away from the ice front, and (iii) in the south.

We build on a number of published sources of bathymetric data with new observations from four recent field campaigns. The existing bathymetric data used in the gridding process here (Figure 1, blue dots) are derived from a targeted seismic bathymetry survey, seismic refraction experiments and drill site measurements (Brisbourne et al., 2014; Nicholls et al., 2012). The depth to grounded ice and known offshore bathymetry of Bedmap2 is included in the gridding process (Fretwell et al., 2013). Surface elevation and ice thickness measurements at Bawden Ice Rise (BIR) are also included (Holland et al., 2015). Here, we integrate these existing data with the new measurements of seabed depth. All data are then gridded to obtain a new bathymetry map of LCIS. We recognise that users of these data are likely to create bathymetric grids using their preferred gridding method, specific to their preferred model and resolution. As such, the gridded product presented here is presented as an aid to discussion and to highlight the value of these new data.

## 3 Data acquisition and processing of new observations

### 3.1 Data Acquisition

In December 2016, 14 seismic bathymetry measurements were made across LCIS, targeting areas of sparse data coverage (Figure 1, magenta dots). The seismic source consisted of a sledgehammer with a plate stamped into the snow surface, or dug down to a shallow ice layer, to improve source consistency of the shots at that location for stacking purposes. Twenty-four Georod receivers (Voigt et al., 2013) were buried to 0.3 m depth, at 10 m spacing, and with a 30 m offset between the shot and

the first receiver. Burying sensors in this way ensures good coupling and provides protection from wind-induced noise. Georods consist of four geophone elements in series, which improves the signal to noise ratio. We recorded 2 s records at 0.125 ms sample interval with a 24-channel data logger. At each site, ~20 hammer blows were recorded using a geophone adjacent to the hammer plate to initiate recording. An additional stack of 10 hammer blows was also recorded for on-site

evaluation of the seismic reflection strength. To determine an accurate surface elevation a dual-frequency GPS system was deployed for the duration of the seismic acquisition at each site.

These data are supplemented by bathymetry measurements from an additional 16 seismic refraction and reflection surveys across LCIS (Figure 1 – orange, black and yellow dots). Although many of these experiments targeted depth profiles of the

firn, the data are suitable for ice-shelf thickness and seabed-depth measurement. The acquisition procedure is similar to that described above and therefore data quality and uncertainties are similar. Details of the acquisition parameters for each experiment are presented in Table 1 with further details in the metadata of each data archive.

Figure 2 presents an example of a seismic gather formed of 10 hammer blows stacked during acquisition. Clear ice-base and

seabed arrivals, as well as multiples thereof, are observed. Where necessary, to help identify reflections, a frequency-wavenumber filter was used to suppress groundroll that may mask the ice-base reflection. An automatic gain control filter and semblance analysis was also used as required to identify arrivals. However, ice-base and seabed reflection traveltimes were measured on raw seismic records, even if a filter was required to help identify arrivals. A relatively thin ice shelf will result in the ice-base reflection arriving within groundroll noise (see Figure 2). In these cases, surface multiples of the ice-base reflection

were used to calculate the primary two-way traveltime through the ice column.

## 3.2 Seismic velocities in ice and water and thickness measurement

We follow the procedures outlined in Brisbourne et al. (2014) to convert from traveltime to thickness. Values of seismic velocity are required to convert traveltimes to layer thickness or depth. A mean seismic velocity in the water column of 1445

$\pm$ 1 m s$^{-1}$ was derived during conductivity-temperature-depth (CTD) measurements made beneath northern and southern LCIS by Nicholls et al. (2012).

The seismic velocity profile in the upper 100 m of the ice shelf, which includes the firn, was measured using the shallow refraction experiments presented here (see Table 1). At each of the refraction sites, a series of surface shots was recorded with

increasing receiver spacing. The first arrivals were picked and converted to a velocity-depth profile using the method described by Kirchner and Bentley (1990). This method relies on a monotonic increase in velocity with depth, an assumption that is supported by observations of smoothly varying traveltimes. Maximum velocities at 100 m depth calculated by inversion of the refraction measurements range from 3698 to 3916 m s$^{-1}$. Below 100 m depth, we assume that ice density is constant and seismic

velocity depends on ice temperature alone. CTD measurements of Nicholls et al. (2012) indicate an ice-base temperature of -2° C. Temperature measurements within the ice column indicate an approximately linear temperature profile with a small range (-14° C at 100 m depth to -2° C at the ice base; Nicholls, 2012, unpublished) and therefore using the relationship of Kohnen (1974), a small range of seismic velocities (3800-3827 m s$^{-1}$). Therefore, below 100 m we linearly interpolate between the velocity measured by seismic refraction at 100 m depth and an ice-base velocity calculated from the temperature-velocity relationship of Kohnen (1974). Where a bathymetry measurement and seismic refraction experiment are not coincident, results from the closest seismic refraction experiment are used to determine ice thickness.

Measurement of the surface elevation allows for the estimation of ice thickness assuming freely floating ice. These estimates can guide the identification of ice-base reflections in the data. The EIGEN-GL04C geoid level (Forste et al., 2008) is removed from the elevation and an empirical relationship determined by Brisbourne et al. (2014) used to calculate ice thickness: geoid-corrected height, $h = (0.113\pm0.005)H + (5.003\pm1.525)$, where H is ice column thickness in metres. This relationship accounts for firn thickness, which affects mean density. The absence of a clear ice-base reflection is not necessarily a result of poor quality data. Under certain conditions, particularly in ice-shelf suture zones, poorly consolidated marine-ice at the base of the ice shelf may result in a weak or absent seismic reflection. At site PRHB04 (see Table 2), in the absence of a clear ice-base reflection, we calculate the ice thickness and its uncertainty from the surface elevation using the empirical relationship described above.

### 3.3 Uncertainties

Errors in picking reflections, seismic velocities and seabed topography all contribute to uncertainties in ice and water-column thickness calculations. Picking of ice and seabed reflections was repeated three times for each site in order to quantify the error, indicating a maximum picking error in the seismic reflections of 0.5 ms. Ice column thickness is determined by the difference in seabed and ice-base arrival times and therefore has an uncertainty of 1.0 ms. We assume a conservative estimate of the uncertainties in seismic velocity in the ice column of 30 m s$^{-1}$ (Kirchner and Bentley, 1990; Rosier et al., 2018). The presence of marine ice in suture zones (Kulessa et al., 2019) or significant warm refrozen ice within the firn column (Hubbard et al., 2016; Ashmore et al., 2017) may result in seismic velocities which deviate from the standard model and introduce greater uncertainty in measured velocities. However, a previous study highlighted the consistency between seismically-derived ice-thickness measurements and those from surface-elevation measurements (Brisbourne et al., 2014). Importantly, where an ice-base reflection can be identified the calculated thickness of the water column is independent of the ice velocity-depth profile used.

Ice-base and seabed topography can introduce additional uncertainty to thickness measurements (Nost, 2004). Calculations of ice and water-column thickness from traveltimes assumes that reflectors are planar and horizontal. Such a geometry results in

a characteristic curvature, or moveout, of traveltimes with increasing receiver offset. Assuming an isotropic seismic velocity structure, any deviation from standard moveout is indicative of dip at the reflecting interface. Brisbourne et al. (2014) used observed deviations from standard moveout to demonstrate that topography across LCIS causes a maximum error in seabed depth of <10 m. However, a full assessment of the error introduced by bed topography would require multiple measurements

at each site at different angles across the slope and this uncertainty is therefore not included here.

We calculate seabed depth by removing ice and water-cavity thickness from surface-elevation data. Uncertainties in dual-frequency GPS elevation measurements are up to ±40 mm. Where direct surface-elevation measurements are not available (dual-frequency GPS measurements not made, see Table 1) the REMA surface DEM of 2017 at 8 m resolution is used (Howat

et al., 2019), resulting in an absolute elevation uncertainty in these areas of ± 2 m. No tidal correction is made to surface elevations, resulting in additional uncertainty of ± 2 m (Brisbourne et al., 2014).

Based on the above uncertainty sources, but excluding the unknown seabed slope, we calculate uncertainties in seabed depths at each site and present these with the seabed depths in Table 1. Uncertainty at site PRHB04, where no ice-base reflection was

observed, has been calculated using the range of ice thickness values indicated by the empirical surface elevation relationship of Brisbourne et al. (2014) and the resultant uncertainty in water column thickness.

## 4 Bathymetry gridding and results

We include a one-kilometre horizontal resolution bathymetry grid of LCIS's cavity in the supplementary material. To produce this grid we interpolated between all existing and new seabed depth measurements. We augmented these data with the

grounding line position, grounded-ice bed depths and offshore bathymetry derived from Bedmap2 (Fretwell et al., 2013). The measured bed geometry of Bawden Ice Rise was derived from Holland et al. (2015) (see Table A1). We use a natural neighbour interpolation implemented in the griddata function of MATLAB (Release 2019a), which is well-suited to a dataset with an uneven distribution of data points (Sibson, 1981). Importantly, the fit to these points does not 'overshoot', which would result in interpolated values that are higher or lower than known values. A weakness of this method is that where seabed topography

changes rapidly with respect to data coverage the seabed may not be well constrained. This results in a discrepancy where the interpolated seabed depth is shallower than the ice draft as reported in Bedmap2. Therefore, in the gridded product we deepen the seabed where this discrepancy occurs and assign a seabed depth to be equivalent to the Bedmap2 ice draft plus a minimum water column thickness of 10 m. This ensures that all interpolated seabed depths are consistent with the Bedmap2 ice thickness.

Figure 3 presents a map of seabed elevation in the LCIS region, resulting from gridding of all available data as described above and with a minimum cavity thickness of 10 m when compared to Bedmap2 ice draft. The analysis and application of such grids is of course dependent on the limitations of data coverage and the gridding method used. Our method does not superimpose

any additional constraints on the resultant bathymetry, as highlighted by the bullseye deepening around the single data point to the south of the Kenyon Peninsula that in reality may form a linear trough. However, the sparse data coverage in that area precludes any definitive knowledge of bed geometry and any gridded product will require careful interpretation.

Figure 4 presents the difference between the grid corrected for Bedmap2 ice draft and the original interpolated grid, highlighting the areas of the cavity where the data coverage and interpolation method are least reliable. In general, this discrepancy occurs where topography is changing rapidly with respect to data coverage, such as along the grounding line. To avoid issues such as this, other methods of interpolation may be preferred. For example, knowledge of past ice flow may be used to prescribe channels around isolated data points, or onshore slopes may be continued into the cavity. Similarly, the

prescribed minimum cavity thickness of 10 m may be adapted to the resolution and type of oceanographic model used. This gridded product is therefore not viewed as a definitive bathymetry for use by the oceanographic community but is used here to highlight the value of these new data. No matter what method is applied, there are intrinsic weakness of gridding with interpolation in regions of sparse data coverage, and by their nature, uncertainties cannot be quantified readily.

**5 Discussion and significance of the data set**

A number of key features that will influence tidal and oceanic circulation through the sub-shelf cavity, and thus affect basal melt rates and melt water circulation are apparent: (1) A relatively deep seabed surrounds Bawden Ice Rise, a key pinning point of LCIS, where LCIS is closest to floatation (Holland et al., 2015). The bathymetry of this area therefore plays a key role in the ice shelf's future stability. A deep seabed here may alter the strong tidal currents that are thought to induce melt in this

region (Mueller et al., 2012) and also reduce the likelihood of re-grounding following any slight thickening. (2) The southern trough, to the north of the Kenyon Peninsula, extends from Mobiloil Inlet to the ice front. Nicholls et al. (2012) highlighted this deepening in southern LCIS as a potential conduit for High Salinity Shelf Water (HSSW) that may access the deeper ice at the grounding line, providing vigorous melting. Similarly, the updated bathymetry also confirms that the Jason Trough in the north also continues through to the open ocean, to the north of Bawden Ice Rise. (3) The sub-shelf cavity to the southeast

of the Kenyon Peninsula is relatively deep. Again, Nicholls et al. (2012) highlight this location as potentially important to the supply of HSSW that sustains melt at the grounding line. (4) All additional point measurements confirm that the where sampled the sub-shelf cavity is particularly deep close to the grounding line between Mobiloil Inlet and the Cole Peninsula. Sub-shelf circulation models highlight that the grounding line, where shelf ice is thickest and therefore deepest, provides a key site for basal melt (Mueller et al., 2012). However, the inconsistency between the interpolated grid and Bedmap2 bathymetry at the

grounding line (Figure 4) highlights the remaining shortcomings of the bathymetric data close to the grounding line.

These data provide a valuable product for the study of ice-ocean interaction beneath LCIS. The interaction of these newly determined cavity features with the sub-shelf circulation pattern requires detailed oceanic modelling to ascertain their importance. The updated bathymetry is a prerequisite to estimating the contribution of sub-shelf melt to thinning of the ice shelf and the contribution of that melt to the global ocean system. Previous studies have highlighted the importance of accurate

bathymetry beneath LCIS, but until now have lacked information on the major troughs delineated by these data. The provision of the spot measurements will allow users to re-grid using other algorithms if required, and allow for rapid assimilation of any new data points that become available in the future. This is not a definitive data set and additional data points that address gaps in the current coverage will always be of value to reduce uncertainty where interpolation has been necessary. As the resolution of ocean models improves, the requirement for greater certainty with regards small-scale features will also increase.

**Data Availability**

The underlying seismic data sets which were used to determine bed depth and ice thickness are available at https://doi.org/10.5285/315740B1-A7B9-4CF0-9521-86F046E33E9A (Brisbourne et al., 2019), https://doi.org/10.5285/5D63777D-B375-4791-918F-9A5527093298 (Booth, 2019), https://doi.org/10.5285/FFF8AFEE-4978-495E-9210-120872983A8D (Kulessa and Bevan, 2019) and https://doi.org/10.5285/147BAF64-B9AF-4A97-8091-

26AEC0D3C0BB (Booth et al., 2019).

**Author contribution**

PRH and AMB led the NERC BAS bathymetry experiment. BK, AJL and ADB respectively led NERC projects SOLIS, MIDAS and RACE. TH, BK, SB, DA, BH, AJL, ADB, EP and JW were involved in field data acquisition. AMB and BK wrote the manuscript with contributions from others. LH and AMB gridded the final bathymetry product.


**Competing interests**

The authors declare no competing interests.

**Acknowledgements**

We acknowledge support by UK Natural Environment Research Council (NERC), the British Antarctic Survey Polar Science

for Planet Earth Programme and NERC grants NE/E012914/1 (SOLIS), NE/L005409/1 (MIDAS) and NE/R012334/1 (RACE). We thank BAS Operations for support of all data acquisition presented here. NERC Geophysical Equipment Facility supplied instruments for the fieldwork under loans 863, 864, 865, 1028 and 1060. The MODIS image from 2018 was retrieved on 2019_10_01 from Earth Science Data and Information System (ESDIS) Project, Earth Science Projects Division (ESPD),

Flight Projects Directorate, Goddard Space Flight Center (GSFC) National Aeronautics and Space Administration (NASA) 2019. We are grateful to Emma Smith, Coen Hofstede and two anonymous reviewers for their constructive comments which helped improve this manuscript.

**Figures**

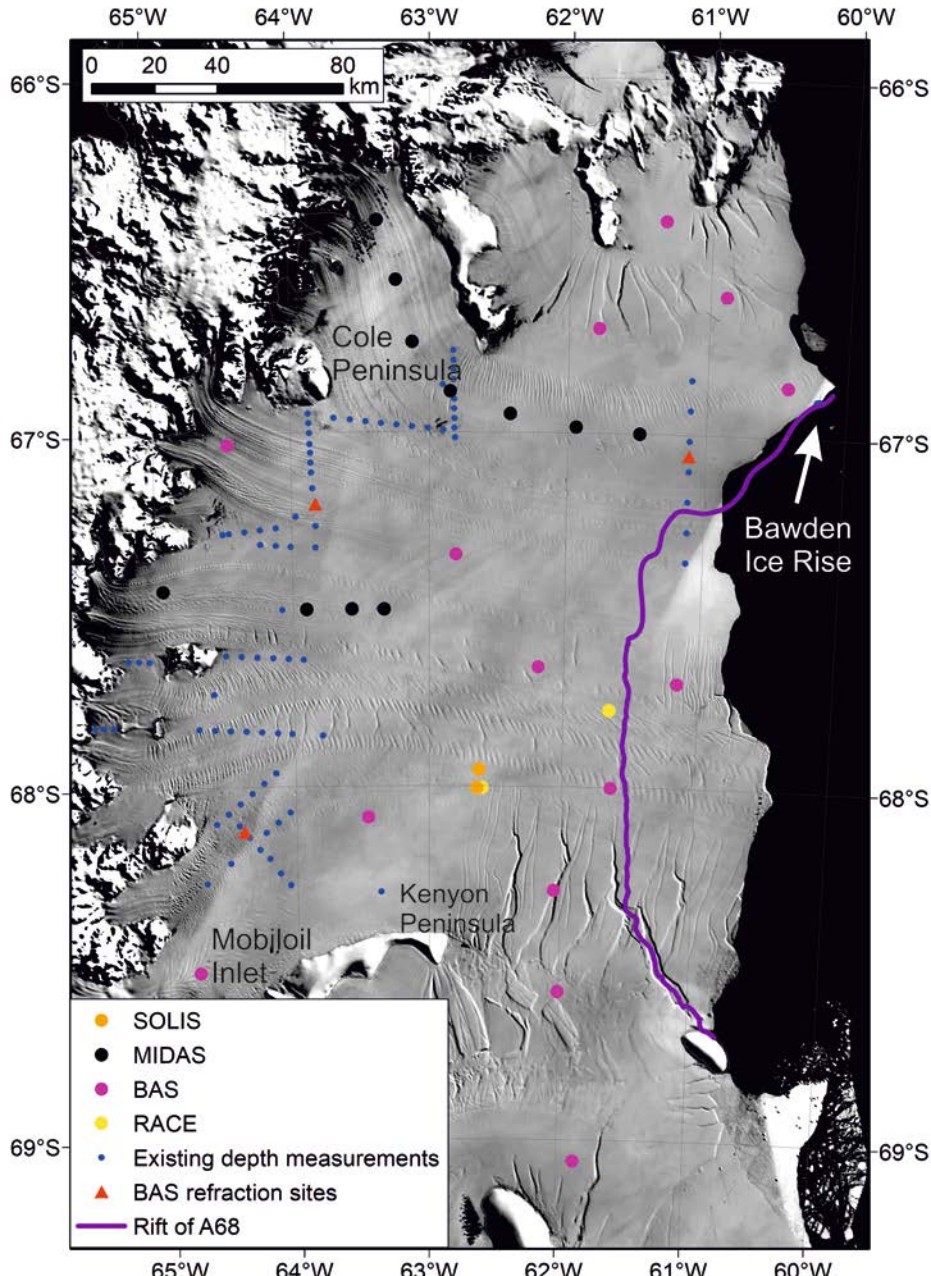

**Figure 1** Map of seismic points used in the gridded bathymetry product of this study. The approximate path of the ice-shelf rift which resulted in the calving of iceberg A68 is highlighted (Jansen et al., 2015). The background is MODIS imagery (Scambos et al., 2007), pre-dating the break-off of iceberg A68 along the rift.

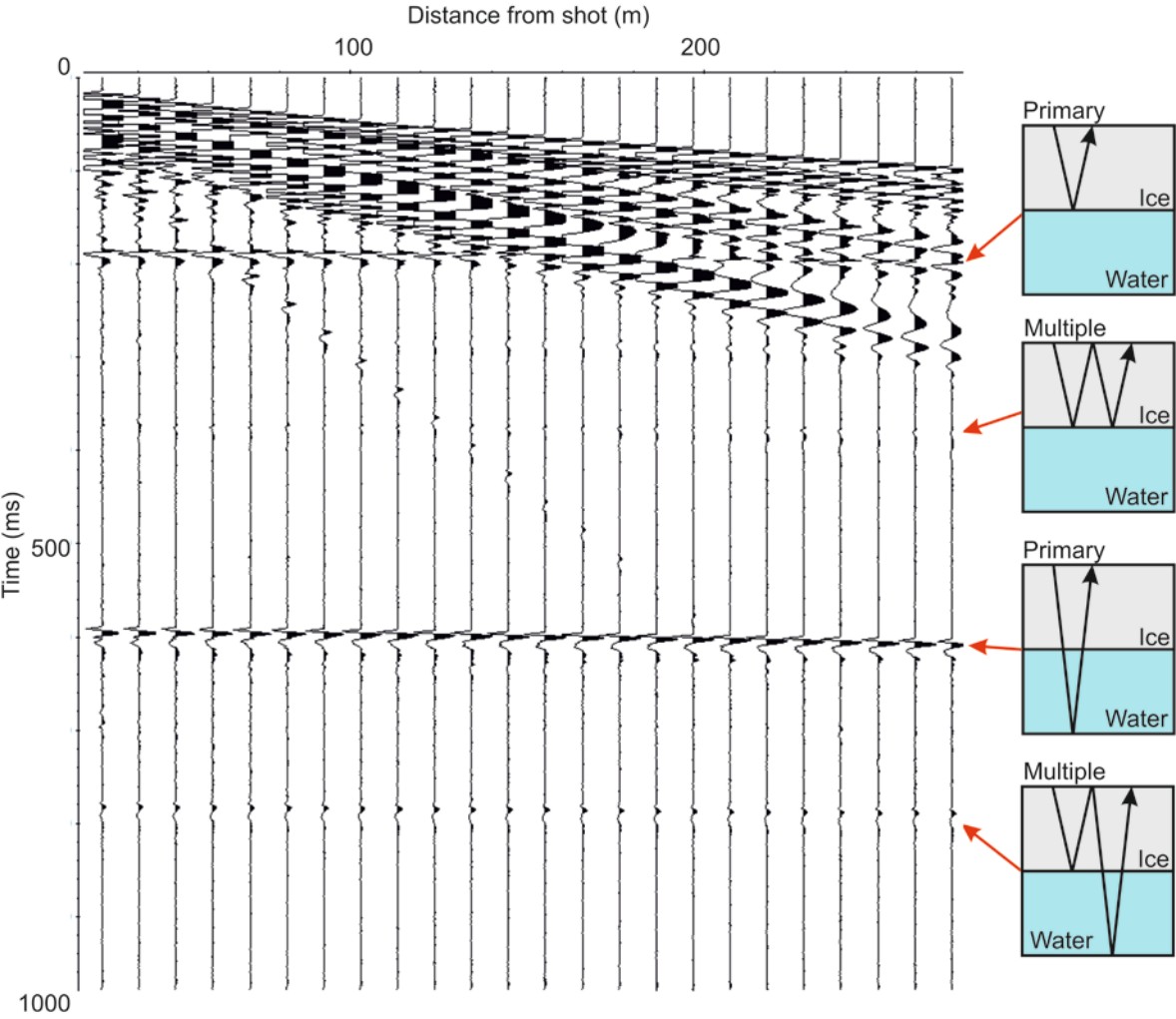

**Figure 2** Example hammer and plate seismic shot gather with readily identified primary seismic reflections and multiples. The primary ice-base reflection at 190 ms is masked by groundroll signal at far offsets.

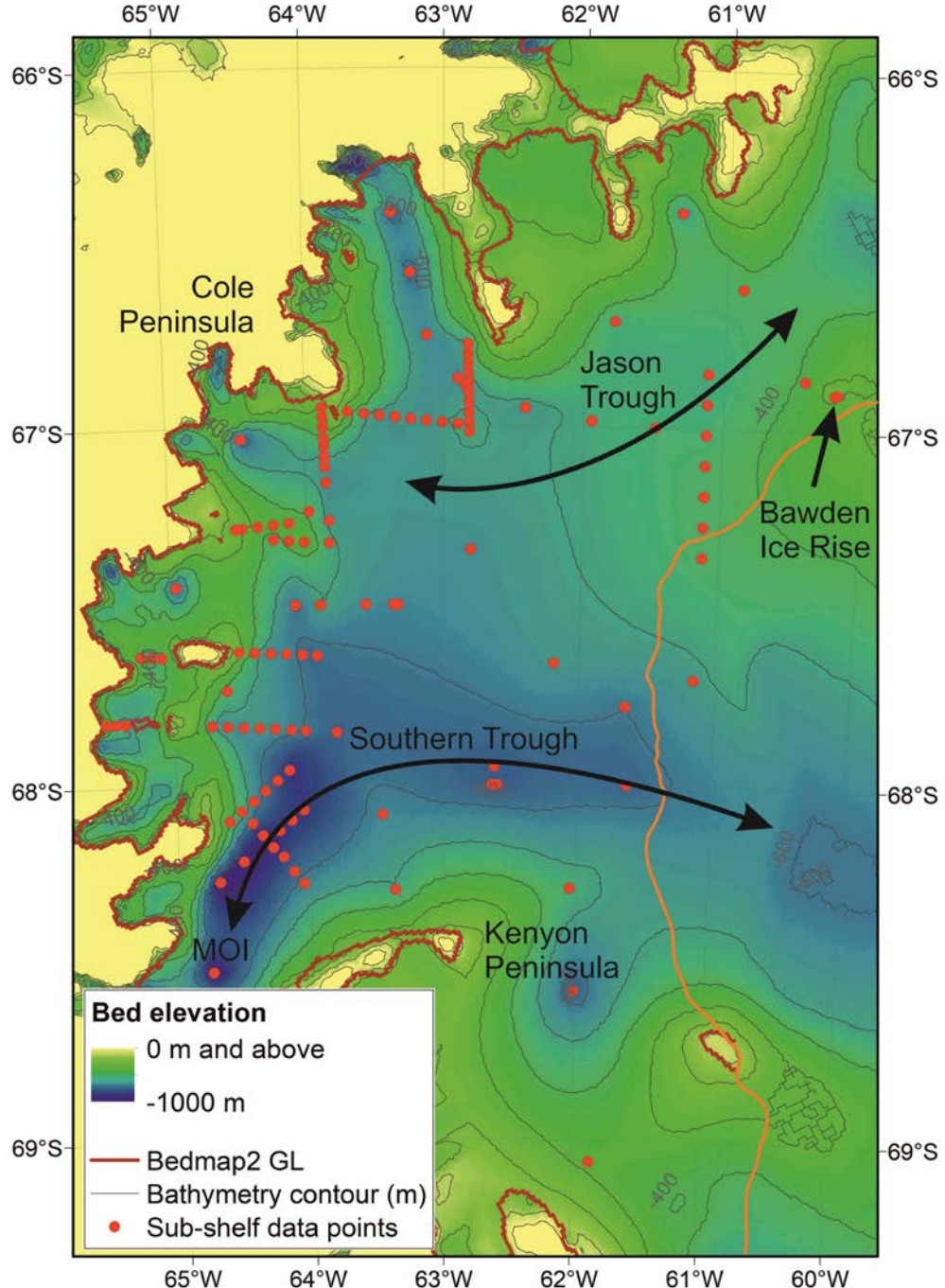

**Figure 3** Updated bathymetry map of Larsen C Ice Shelf with large-scale features highlighted. For clarity, elevations above 0 m are unscaled and we label only the -400 and -600 m contours. The brown line represents the Bedmap2 grounding line (GL) (Fretwell et al., 2013). The orange line represents the ice front on 20th December 2018 highlighting the new ice front following the calving of iceberg A68 determined from MODIS imagery (Vermote and Wolfe, 2015). MOI – Mobiloil Inlet. The black arrows highlight likely oceanographic conduits as discussed in the text.

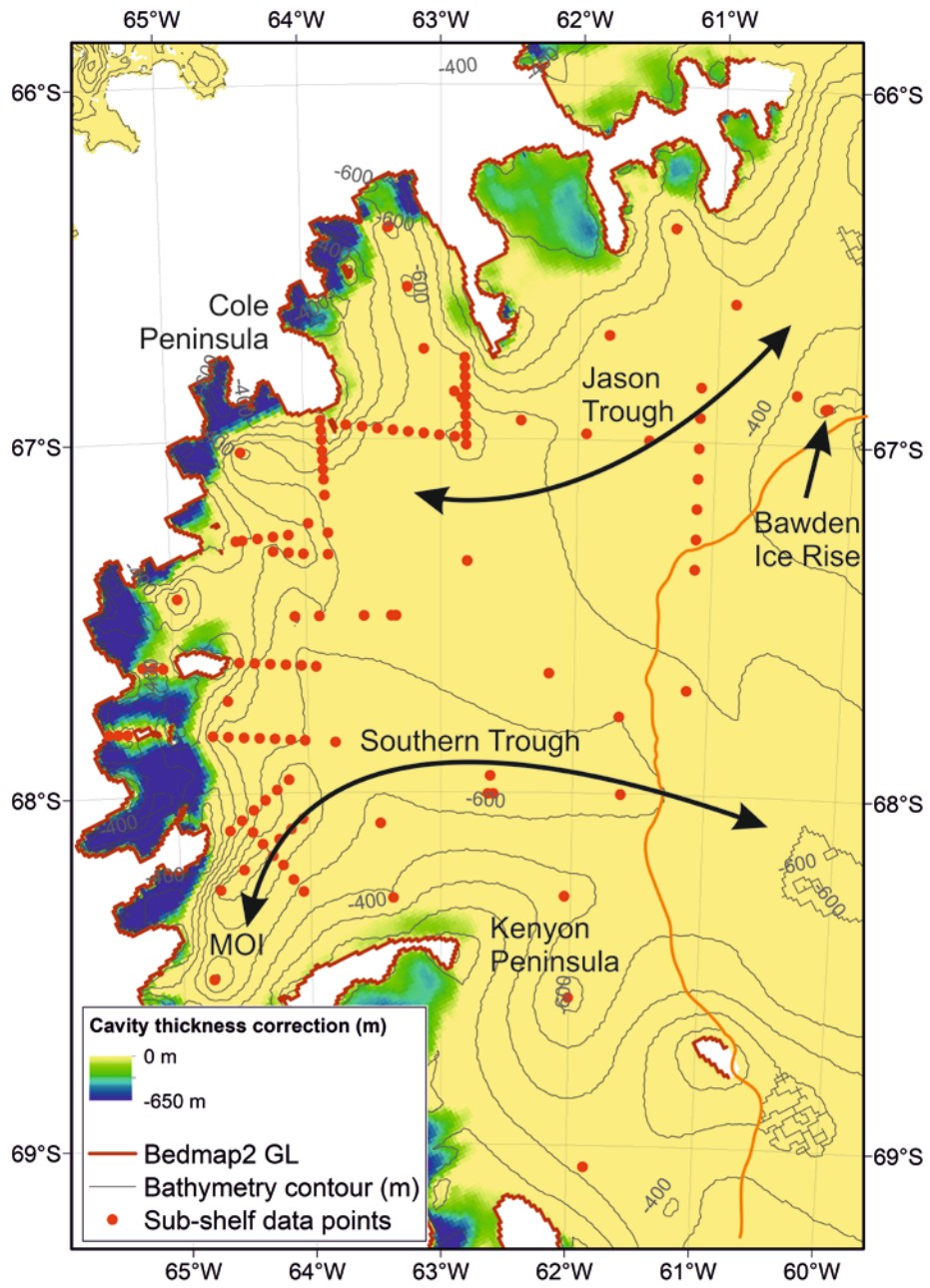

**Figure 4** Difference between the grid corrected for the Bedmap2 ice draft mismatch and the original natural neighbour interpolation grid in metres, highlighting areas where the cavity has been forced to 10 m.

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

**Table 1** Field acquisition parameters for new data presented here. *Note: Due to the use of a range of acquisition geometries, the specific geometry used at every refraction experiment site is included in the data repository. BAS Refraction data were published previously in Brisbourne et al. (2014) but are again presented here as they form part of the analysis of new data.

| Acquisition parameter | BAS Bathymetry | BAS Refraction | MIDAS Refraction | SOLIS Refraction | RACE Reflection |
|---|---|---|---|---|---|
| Source type | Hammer | Pentolite (surface) | Hammer | Pentolite (1 m depth) | Hammer |
| Trigger type | Uphole geophone | Blaster initiated | Impact-sensitive switch | Blaster initiated | Impact-sensitive switch |
| Receiver type | Georod | Georod | Geophone | Geophone | Geophone |
| Receiver corner frequency | 40 Hz | 40 Hz | 100 Hz | 100 Hz | 10 Hz |
| Receiver spacing | 10 m; 30 m offset to first receiver | 2.5m to 10m; 5m to 30m; 10m thereafter | 48 channels increasing from 0.5 to 10m* | 2.5m to 10m; 5m to 30m; 10m thereafter | 10 m |
| Maximum offset (m) | 260 | 610 | 1110 | 1110 | 330 |
| Sample interval (ms) | 0.125 | 0.125 | 0.0625 | 0.0625 | 0.0625 |
| Record length (s) | 2 | 2 | 1 | 1 | 1 |

**Table 2** Location and seabed depth measurements and associated uncertainty of all new points used in this study.

| SITE | Project | Latitude (°) | Longitude (°) | Elevation (m) | Ice shelf thickness (m) | Water column thickness (m) | Seabed elevation (m) | Seabed depth uncertainty (m) |
|---|---|---|---|---|---|---|---|---|
| SLGS | SOLIS | -68.005 | -62.642 | 55.00 | 302.4 | 410.4 | -657.8 | 8.2 |
| SLGN | SOLIS | -67.954 | -62.624 | 53.00 | 300.8 | 410.4 | -658.1 | 8.2 |
| CI-0-wet | MIDAS | -66.403 | -63.376 | 76.87 | 559.3 | 176.5 | -659.0 | 16.8 |
| CI-0-dry | MIDAS | -66.402 | -63.371 | 70.62 | 577.3 | 173.4 | -680.1 | 17.1 |
| CI-20 | MIDAS | -66.571 | -63.238 | 66.73 | 499.3 | 213.8 | -646.4 | 14.8 |
| CI-40 | MIDAS | -66.746 | -63.121 | 56.21 | 439.9 | 192.2 | -575.9 | 15.9 |
| CI-60 | MIDAS | -66.885 | -62.847 | 49.74 | 366.4 | 222.2 | -538.9 | 14.4 |
| CI-80 | MIDAS | -66.948 | -62.415 | 48.05 | 301.2 | 282.5 | -535.6 | 12.4 |
| CI-100 | MIDAS | -66.984 | -61.939 | 48.05 | 277.9 | 243.2 | -473.0 | 13.6 |
| CI-120 | MIDAS | -67.000 | -61.481 | 47.21 | 262.2 | 237.5 | -452.5 | 13.8 |
| WI-70 | MIDAS | -67.500 | -63.336 | 49.00 | 297.6 | 326.4 | -575.0 | 11.4 |
| WI-60 | MIDAS | -67.500 | -63.569 | 49.65 | 283.9 | 324.0 | -558.2 | 11.5 |
| WI-45 | MIDAS | -67.500 | -63.901 | 49.70 | 303.0 | 242.0 | -495.3 | 13.6 |
| WI-00 | MIDAS | -67.444 | -64.953 | 59.10 | 445.8 | 254.9 | -641.6 | 13.2 |
| PRHA01 | BAS | -67.346 | -62.803 | 52.44 | 282.5 | 317.7 | -547.7 | 9.6 |
| PRHA02 | BAS | -67.662 | -62.189 | 51.24 | 267.4 | 312.0 | -528.2 | 9.8 |
| PRHA03 | BAS | -66.609 | -60.884 | 46.00 | 211.3 | 262.2 | -427.5 | 11.0 |
| PRHA04 | BAS | -66.705 | -61.785 | 47.19 | 236.3 | 280.1 | -469.2 | 10.5 |
| PRHA05 | BAS | -68.294 | -62.048 | 48.15 | 236.5 | 269.7 | -458.1 | 10.8 |
| PRHA07 | BAS | -66.860 | -60.424 | 43.83 | 235.1 | 200.3 | -391.5 | 13.4 |
| PRHB01 | BAS | -68.525 | -64.761 | 78.08 | 547.9 | 382.4 | -852.2 | 8.6 |
| PRHB02 | BAS | -68.088 | -63.458 | 58.32 | 329.1 | 293.2 | -564.0 | 10.2 |
| PRHB03 | BAS | -68.002 | -61.634 | 48.95 | 266.8 | 417.0 | -634.9 | 8.1 |
| PRHB04 | BAS | -68.582 | -62.006 | 27.55 | 130.0 | 551.4 | -653.8 | 31.7 |
| PRHB05 | BAS | -69.062 | -61.864 | 38.87 | 146.2 | 224.0 | -331.3 | 12.3 |
| PRHB06 | BAS | -67.034 | -64.460 | 48.83 | 277.4 | 426.3 | -654.9 | 8.0 |
| PRHB12 | BAS | -67.705 | -61.156 | 47.72 | 228.1 | 338.2 | -518.6 | 9.2 |
| PRHB15 | BAS | -66.400 | -61.328 | 51.54 | 277.6 | 307.7 | -533.7 | 9.9 |
| RACE-S1 | RACE | -67.783 | -61.657 | 50.35 | 274.9 | 375.1 | -599.7 | 10.6 |
| RACE-S2 | RACE | -68.005 | -62.600 | 53.65 | 297.9 | 413.1 | -657.3 | 10.1 |

## Appendix A

**Table A1 Previously published seabed depth measurements included in the gridding process (Holland et al., 2015; Brisbourne et al., 2014)**

| Latitude (°) | Longitude (°) | Depth (m) | Latitude (°) | Longitude (°) | Depth (m) | Latitude (°) | Longitude (°) | Depth (m) |
|---|---|---|---|---|---|---|---|---|
| -67.500 | -64.083 | -615.9 | -67.286 | -64.508 | -327.0 | -68.102 | -64.140 | -758.8 |
| -67.500 | -63.366 | -577.6 | -66.947 | -63.872 | -334.3 | -68.074 | -64.049 | -748.7 |
| -67.017 | -62.813 | -479.2 | -66.975 | -63.869 | -341.8 | -68.104 | -64.607 | -428.7 |
| -66.990 | -62.816 | -483.6 | -67.002 | -63.867 | -341.2 | -68.076 | -64.515 | -449.3 |
| -66.962 | -62.817 | -481.3 | -67.033 | -63.863 | -414.0 | -68.048 | -64.424 | -572.7 |
| -66.935 | -62.818 | -506.9 | -67.058 | -63.861 | -420.8 | -68.019 | -64.332 | -614.7 |
| -66.907 | -62.820 | -523.4 | -67.086 | -63.858 | -501.7 | -67.991 | -64.242 | -703.6 |
| -66.880 | -62.821 | -529.1 | -67.114 | -63.855 | -530.6 | -67.962 | -64.151 | -768.8 |
| -66.853 | -62.823 | -493.5 | -67.158 | -63.849 | -555.0 | -68.109 | -64.435 | -639.6 |
| -66.826 | -62.824 | -474.8 | -67.265 | -63.829 | -528.3 | -68.143 | -64.360 | -799.6 |
| -66.798 | -62.825 | -494.2 | -67.326 | -63.830 | -481.6 | -68.177 | -64.283 | -794.3 |
| -66.771 | -62.827 | -513.1 | -67.642 | -63.930 | -620.6 | -68.204 | -64.206 | -702.5 |
| -66.960 | -63.689 | -382.6 | -67.639 | -64.041 | -650.6 | -68.246 | -64.129 | -621.2 |
| -66.965 | -63.570 | -500.7 | -67.637 | -64.156 | -534.9 | -68.280 | -64.053 | -577.5 |
| -66.971 | -63.459 | -546.1 | -67.634 | -64.273 | -585.1 | -66.894 | -60.193 | -124.8 |
| -66.976 | -63.346 | -541.6 | -67.631 | -64.389 | -477.5 | -66.894 | -60.194 | -122.2 |
| -66.980 | -63.234 | -522.7 | -67.629 | -64.505 | -236.0 | -66.894 | -60.195 | -129.7 |
| -66.985 | -63.122 | -557.5 | -67.857 | -63.794 | -641.0 | -66.894 | -60.196 | -123.6 |
| -66.990 | -63.009 | -498.8 | -67.852 | -64.023 | -599.8 | -66.894 | -60.197 | -124.7 |
| -66.994 | -62.897 | -492.9 | -67.849 | -64.131 | -586.2 | -66.894 | -60.198 | -123.9 |
| -66.847 | -61.114 | -479.7 | -67.846 | -64.249 | -539.9 | -66.895 | -60.199 | -119.2 |
| -66.932 | -61.116 | -466.2 | -67.843 | -64.366 | -540.1 | -66.895 | -60.200 | -126.6 |
| -67.018 | -61.117 | -454.0 | -67.841 | -64.484 | -531.8 | -66.895 | -60.201 | -126.7 |
| -67.104 | -61.119 | -434.0 | -67.838 | -64.601 | -547.7 | -66.895 | -60.202 | -125.1 |
| -67.190 | -61.120 | -434.0 | -67.835 | -64.718 | -532.0 | -66.895 | -60.203 | -123.9 |
| -67.276 | -61.121 | -424.6 | -67.737 | -64.599 | -448.5 | -66.895 | -60.204 | -123.2 |
| -67.362 | -61.123 | -432.7 | -67.636 | -65.229 | -496.6 | -66.895 | -60.204 | -122.5 |
| -67.323 | -64.012 | -316.0 | -67.638 | -65.153 | -481.5 | -66.895 | -60.205 | -120.3 |
| -67.320 | -64.120 | -321.3 | -67.640 | -65.075 | -490.6 | -66.895 | -60.206 | -128.1 |
| -67.316 | -64.235 | -461.1 | -67.823 | -65.502 | -392.3 | -66.895 | -60.207 | -126.6 |
| -67.238 | -63.973 | -413.1 | -67.823 | -65.431 | -567.8 | -66.895 | -60.208 | -127.2 |
| -67.270 | -64.120 | -358.9 | -67.823 | -65.361 | -625.6 | -66.896 | -60.209 | -127.7 |
| -67.275 | -64.234 | -328.2 | -68.272 | -64.692 | -868.1 | -66.896 | -60.210 | -130.7 |
| -67.279 | -64.348 | -314.3 | -68.216 | -64.507 | -856.2 | -66.896 | -60.211 | -133.6 |
| -67.284 | -64.462 | -306.3 | -68.130 | -64.231 | -763.4 | -66.896 | -60.212 | -136.7 |
|  |  |  |  |  |  | -66.896 | -60.213 | -137.5 |

