# Peer review of "An updated seabed bathymetry beneath Larsen C Ice Shelf, west Antarctica"

_Earth System Science Data, 2019_

## Referee Comment (RC1) · Emma C. Smith (Referee) · 14 Nov 2019

**General Comments:**

This study presents a new gridded bathymetry of the ice-shelf cavity beneath Larsen C Ice Shelf, complied from new and existing seismic data sets and some existing drill-site measurements. The article is concise and well written covering all relevant aspects of data processing. This data set presents a significant improvement on previously available cavity bathymetry. As identified by the authors, this data set will be of use in improving predictive models of the future evolution of the potentially vulnerable Larsen C Ice Shelf. All data sets are present at the links given.

My only major comment is easily remedied and of a technical nature: The labelling of the data sets used is somewhat confusing. It isn't clear to me how the new data described in Section 3.1 is related to Figure 1 and Table 1, I have gone back and forth between them a few times and tried to cross-check, but I am still not entirely sure. My suggestions would be:

1. Adjust the labelling of Figure 1, so it is clear and consistent with Table 1. Add to the legend in Figure 1 to indicate which data sets (e.g. BAS reflection, MIDAS, RACE etc…) are indicated by the different coloured points. For example, I am not clear if the yellow points are referring to just the dedicated bathymetry measurements described at the start of Section 3.1, or also some of the other supplementary data sets (some of which are reflection and some refraction experiments)?
2. Add references in the text of Section 3.1 to Figure 1
3. Add references in the text of Section 3.1 to the different survey names/campaigns given in Table 1 - I have made some suggestions in the specific comments as to where I think this would be useful.
4. Add a row to Table 1 to give the reference to the paper/doi where data can be found.

**Minor/Specific Comments:**

Pg1, L21 – State that "new water column thickness measurements" are from seismic data.

Pg2, L10 – step or steep increase?

Pg2, L21 – Additional references:
*Goldberg, D. N., Gourmelen, N., Kimura, S., Millan, R., & Snow, K. (2019). How Accurately Should We Model Ice Shelf Melt Rates? Geophys. Res. Lett., 46 (1), 189{199. doi: 10.1029/2018GL080383*

*Pattyn, F., Favier, L., Sun, S., & Durand, G. (2017). Progress in Numerical Modeling of Antarctic Ice-Sheet Dynamics. Current Climate Change Reports, 3 (3), 174{184. doi: 10.1007/s40641-017-0069-7*

Pg2, L31 – Rephrase sentence starting "The geometry of LCIS…". I had to read it twice as it sounds like the specific locations were measured by inverting gravity data, rather than the gravity inversions being used to help choose targets for the specific measurements.

Pg3, L5 – Reference to Figure 1 (blue dots)

Pg3, L6 – Add Reference to Nicholls et al., 2012 when boreholes are mentioned.

Pg3, L11 – Consider changing title to "New Data Acquisition for clarity.

Pg3, L12 – Reference to Figure 1 (yellow dots?) – see general comments.

Pg3, L14 – Does digging the plate in "improve source consistency" or coupling? It sounds odd to use "consistency" here, as you have described two different methods of placing the source.

Pg4, L15 – Explain the 30 m offset is between the source and the first geophone. Consider moving this a few sentences later, where you introduce the source, rather than here where you are talking about receivers.

Pg3, L18 – "using a geophone trigger adjacent to the hammer plate" add "to start the recording" or something similar.

"A stack on 10 hammer blow were also…" – I'm not quite clear on what this means? Were 10 of the 20 hammer blows stacked to evaluate reflection strength, or were an additional 10 blows made and stacked on site for evaluation? A little re-phasing needed here, as the sentence seems a bit lost.

Pg3, L22 – See general comments above, I am not clear on where the "supplementary surveys" are on Figure 1.

Pg3, L30 – "constrain arrivals" – I think "identify" would be a better word to use here, as you state that travel times were measured on the raw gathers so semblance and AGC wasn't actually used to constrain them?

Pg4, L1 – Nice idea to use the multiples in these cases!

Pg4, L10 – Are these the "BAS refraction sites" in Figure 1 or all refraction measurements? If so, reference here. As above, some confusions with which data set is which.

Pg4, L26 – Add reference to Table 2, after "At site PRHB4"

Pg5, L29 – "We interpolated all available" change to "We gridded all available" or "We interpolated between all available"

Pg6, L6 – Errors on the gridded product is potentially much larger that the errors quoted in Section 3.3 - a comment to that effect here would be good.

Figure 1: As mentioned above. I am confused with the labelling of survey data here, compared to the text in Section 3.1 and Table 1: Are blue points those from Brisbourne et al., 2014? Are the yellow points a combination of MIDAS, SOLIS and RACE and new

reflection surveys? What are the red points, just BAS refraction or ALL refraction surveys? Please clarify.

Figure 2: Add labels (e.g. P1, M1, P2, M2) next to diagrams on right hand side to signify which are multiples and which primaries – something similar to Brisbourne et al., (2014). It might not be clear to those without a seismic background what they are are looking at.

Table 1: Add column for reference to paper where data is presented, where relevant.

---

## Referee Comment (RC2) · Anonymous Referee #2 · 22 Nov 2019

**General comments**

The authors present new point seismic data collected beneath the Larsen C Ice Shelf in West Antarctica. These new data are used to produce a bathymetric grid of the seabed beneath the ice shelf and some of the main implications of features revealed in this new grid are discussed. This data set is crucial to improving modelling efforts in an important and rapidly changing region, however I find the paper to be very light on important details and I have some concerns about the bathymetric grid that is presented. These major issues are listed below and are followed by more minor technical corrections.

**Specific comments**

- The paper is very light on details and this is a particular weakness in terms of the error estimates and gridding as detailed below, but also throughout the paper in general. For example, I think a section discussing the problems with other estimates of sub-shelf cavity and the difficulties in obtaining these measurements is worthwhile. Section 5 is very brief and could greatly benefit from more careful analysis and discussion, particularly in the context of future ice-shelf stability. Furthermore I think there should be more care to emphasise the weaknesses of the grid and the interpretation that follows from it in areas where only one or two data points are available. Finally, many parts of the paper are completely missing references.

- Section 3.3 seems to rely entirely on analysis made in previous studies and makes no attempt that I can see to constrain uncertainties using the newly collected data. How are the picking errors estimated? The assumption of a linear variation in ice temperature from 100m below the surface to the base of the ice shelf is almost certainly flawed when you consider a typical ice shelf temperature profile. Why is a comparison not made between ice thicknesses obtained through surface and seismic measurements, surely this is easily done and worthwhile since in some cases the former is used rather than the latter. How is the GPS error determined?

- My main concern is with the bathymetric grid itself. The paper in its current form presents this grid, rather than individual point measurements, as the main result. Section 4 that describes the gridding does not go into sufficient detail. What grounded-ice-depth measurements are added into the grid, those from Bedmap2? So the gridded grounding line position is consistent with that dataset? The authors state that the ice draft in Bedmap2 is better constrained, do they mean the draft as calculated from floatation in regions far away from their own measurements? If so the last sentence is wrong since these thickness values in Bedmap2 are not measurements and therefore not 'known'.
  I find the choice of natural neighbour interpolation very puzzling and I think this choice will dramatically affect the resulting grid. I don't think the authors' justification is sufficient, but if there is a very good reason for this choice I would like a discussion and appropriate references. Many deep bathymetric points near the grounding line end up as bullseyes in the grid whereas they are almost certainly located in the troughs of paleo-ice-streams.
  Deepening the seabed to ensure an open cavity in some cases is clearly a necessary evil but how extensively is this done? If this is going to be used by ice sheet modellers then I would strongly suggest a much greater minimum thickness than 10m, since even small transient dhdt values during the start of a simulation will cause the ice shelf to ground and completely change ice flow in the region. An uncertainty estimate would be useful but presumably would be difficult to produce without using a more advanced gridding algorithm (which I strongly feel should be done, or at least a comparison made to justify the choice made here). Also given that many of these data are already published elsewhere and lead to the generation of a previous grid, I think a direct comparison is

essential to be able to ascertain which features are genuinely new discoveries resulting from the data presented here.

Overall, more detailed discussion is needed. I realise a perfect grid is impossible with limited data, but I do not feel that sufficient care has been taken with grid generation and I think the resulting discussion skips over these issues, particularly given that it is presented as the main result of the paper.

**Technical corrections**

Throughout the paper: Hyphenation rules inconsistently applied

p. 1 l. 21: Capitalise West (also in title)

p. 1 l. 28-30 Repeat of earlier parts of the abstract

p.2 l. 3: Missing references for buttressing e.g. Rott 2002, Furst 2016, Reese 2018

p.2 l. 14: Missing references for Antarctic Peninsula warming

p. 2l. 26: accurately predict

p. 3 l. 9: here and elsewhere should be bathymetric when used as adjective and bathymetry when a noun

p. 3 l. 16 Ensure that units are in-line with numbers

p. 5 l. 1: How were picking errors determined, was a repeat of each pick done?

p. 5 l. 22: What is the vertical GPS accuracy based on, why were some surface elevation measurements not available?

p.5 l. 26: Based on all of the above uncertainty sources, we arrive at a cumulative overall uncertainty…

p. 6 ;. 12: Surely another crucial aspect of the bathymetry around this ice rise is whether slight thickening could lead to extensive re-grounding.

p. 6 l. 16-17: I don't see this path of Jason trough north of Bawden Ice Rise since the bathymetry there is shallower than the route south.

p.6 l. 17-19: This is based on one data point and the gridding here has just created a presumably unrealistic bullseye around that point. Presumably the reality is that there is a trough here leading towards the grounding line that is missed in the grid and this should be discussed.

p. 6 l. 21: References needed.

Fig. 3: Given that this is the main figure of the paper I think it needs a lot more work. Firstly the resolution is far too low for publication. The background MODIS imagery is either completely missing or not discernible. Please highlight the grounding line by making it a bolder contour. What do the black arrows indicate? Add label for Mobiloil inlet and Cole Peninsula which are discussed in the discussion text.

---

## Referee Comment (RC3) · Anonymous Referee #3 · 26 Nov 2019

This study presents new bathymetric information derived from seismic shots beneath the Larsen-C Ice cover. The Significance of new information has no doubt concerning ocean circulation and global climatological issues. However, it is not very clear to me up to which level your new data improve previous datasets. The main global comment that I may have from reading the paper several time is that further care should be taken on making the difference on the contribution of the existing data and the motivation/input from the new data. Specific comments - Location of previous work: I believe that a table giving summary statistics of the previous dataset would be valuable (columns could be like: survey name, survey date, type/sensor, number of measurements, estimated vertical accuracy, estimated horizontal accuracy. - Using the table proposed above you can detail the new dataset - It is not very clear to me how the gridding methodology is

done. I understand you've used natural neighbour interpolation. You should provide a schema of your procedure in which we could see the data flow, the different steps (data preparation, gridding process, corrections) and the parameters used. - Your gridding correction steps (line 2-5, p6) is not clear. It looks to me as some sort of data tweaking. Please see reviewers #2 on this point. - Concerning the two last points I believe that the minimal aim of your paper and more specifically section 4 is to enable any readers/data user to be able to reconstruct the bathymetric grid. Therefore I suggest being more explicit in your gridding methodology. Algorithm, implementation, software, parameters . . .

I do not pretend to be able to comment on the English or the style; however I would suggest limiting the vagueness to its minimum. You should be more explicit and limit yourself from using terms like "relatively", "more reliable", "where required", "consistent", "much lower", "further uncertainty", . . .

---

## Referee Comment (RC4) · Coen Hofstede (Referee) · 29 Nov 2019

General comments: The paper presents addition of seismic point measurements to existing ice column–water column thickness measurements at Larsen C Ice Shelf (LCIS). By measuring the travel times of the ice shelf base and seabed, the ice shelf thickness and water column are calculated. In addition to existing data points and bedmap2, a sub-shelf bathymetric map of LCIS is created.

The paper is well written, clearly built-up, and easy to follow. The method to calculate ice thickness and water column is straightforward and well explained, but certain parts are described vaguely,making it hard to judge the quality of the presented data, such as the p-wave velocity of the ice column and the gridding process.

[Figure]

The bathymetric sub-shelf map of LCIS is a valuable addition to the gap in bathymetric data under Antarctic ice shelves. The data points better constrain the sub-shelf bathymetry such as their key findings show, and improve the modeling of the ocean-shelf interaction. LCIS is probably next in line to disintegrate.

Specific Comments: 3.2/3.3 Seismic velocities/Uncertainties The velocity analysis of the firn/ice column is well explained but a number (or range) for the ice velocity(ies) would be nice. Indirectly this is mentioned at the uncertainty of the ice column thickness, being 3.8m at 1 ms uncertainty, which suggests the ice velocity is 3800m/s. If that is the case I come to half of the suggested uncertainty as the times of reflections are TWTs. To understand the uncertainty of the ice column, velocities of the ice column are essential.

It is not clear if the measurements are corrected for tides, I suspect not. This is important for those shots that do not show no ice base return. With a tidal range of 2 m, I would come to approximately 20m inaccuracy. How many shots do not have this ice base return, one at PRHB4 or more?

Although the error analysis is clearly described and the order of magnitude is correct, the choice of 10m accuracy seems somewhat arbitrary to me. Why not 9m or 13m I wonder?

4 Bathymetric gridding I think it is important here to be clear about the gridding method is used rather then "which is well suited to a dataset with an uneven distribution of data point". It is important to know how you get from data points to the gridded bathymetry map. A reference possibly?

I find the phrasing about the gridding problem at places where the "calculated seabed is shallower than …...the ice draft of the Bedmap2 dataset" unclear. Are these calculations ignored or overruled by a deeper seabed? If so it would make sense to mark these data points in map 3 so that we know exactly what data points have been used in the gridding

[Figure]

Figure 1: The text (3.1) and Table 2 mention 30 measurements (14 seismic bathymetry measurements and 16 seismic refraction and reflection surveys). In the figure I see 28 yellow dots (new measurements) and 3 red dots. - How do these 28+3=31 dots relate to the 30 measurements from Table 2? Please explain in the caption or adjust the figure.

Figure 3: Please use another color for the contour lines. They can hardly be made out.

Technical corrections: Table 1, receiver spacing MIDAS: Why an asterisk?
* * *

---

## Author Comment (AC1) · 29 Feb 2020

**An updated seabed bathymetry beneath Larsen C Ice Shelf, west Antarctica"
by Alex Brisbourne et al.**

**Response to reviewers - February 2020**

We are grateful to the four reviewers for their careful analysis of the manuscript. We have endeavoured to address all the issues raised and outline our responses below with reference to the updated manuscript. Our responses are in blue.

**General response**

We recognise that some of the reviewers found the manuscript to be light on detail. We acknowledge that this was due to drawing on previous comprehensive studies with more exhaustive data sets. To this end, we have added more detail for the new data to make this paper a more robust standalone piece of work. However, we have resisted the need to provide too much background detail such that this remains most fundamentally a "Data description paper" and we envisage this being its primary purpose to the oceanographic community. For background to what we originally conceived as the purpose of the article we would refer to the aims of ESSD, outlined here: https://www.earth-system-science-data.net/about/aims_and_scope.html. In particular "Articles in the data section may pertain to the planning, instrumentation, and execution of experiments or collection of data. Any interpretation of data is outside the scope of regular articles. … Any comparison to other methods is beyond the scope of regular articles."

The reviewers raised a number of issues with the gridding methodology. Our experience of collaborating with the oceanographic community is that invariably oceanographers will create bathymetric grids using their preferred method, specific to their preferred model and resolution. As such, they are unlikely to use gridded products directly in their models. The gridded product presented here is simply an aid to discussion and to highlight the value of these new data. However, we have added extra detail to the manuscript to ensure we are entirely clear with our methods and that the grid can be reproduced. We do not however go any further with the evaluation of the gridding method or interpolation as this will be specific to the data users' preferred gridding method and model parameterisation. We do now ensure that we clearly highlight the gridded product as a demonstration of the value of the data points rather than a tool for use directly.

This then leads in to comments about what the new data points introduce to the bathymetry map. In this sense the reviewers are ignoring the value of data where previously there were no measurements. Even if a data point in a previously unmeasured location matches depths previously derived by interpolation, this data point now confirms the validity of the interpolation at that site and users can have confidence in any results for that site. Without this confirmatory data point, interpretations would always be subject to caveats regarding the interpolation method.

**Reviewer #1 Emma Smith**

**General Comments:**
This study presents a new gridded bathymetry of the ice-shelf cavity beneath Larsen C Ice Shelf, complied from new and existing seismic data sets and some existing drill-site measurements. The article is concise and well written covering all relevant aspects of data processing. This data set presents a significant improvement on previously available cavity bathymetry. As identified by the authors, this data set will be of use in improving predictive models of the future evolution of the potentially vulnerable Larsen C Ice Shelf. All data sets are present at the links given.

My only major comment is easily remedied and of a technical nature: The labelling of the data sets used is somewhat confusing. It isn't clear to me how the new data described in Section 3.1 is related to Figure 1 and Table 1, I have gone back and forth between them a few times and tried to cross-check, but I am still not entirely sure. My suggestions would be:

1. Adjust the labelling of Figure 1, so it is clear and consistent with Table 1. Add to the legend in Figure 1 to indicate which data sets (e.g. BAS reflection, MIDAS, RACE etc…) are indicated by the different coloured points. For example, I am not clear if the yellow points are referring to just the dedicated bathymetry measurements described at the start of Section 3.1, or also some of the other supplementary data sets (some of which are reflection and some refraction experiments)?
Done

2. Add references in the text of Section 3.1 to Figure 1
Done

3. Add references in the text of Section 3.1 to the different survey names/campaigns given in Table 1 - I have made some suggestions in the specific comments as to where I think this would be useful. Done as per details below.

4. Add a row to Table 1 to give the reference to the paper/doi where data can be found. As only the BAS data have found their way into a publication previously (hence this paper) this is seen as excessive.

**Minor/Specific Comments:**
Pg1, L21 – State that "new water column thickness measurements" are from seismic data. Done
Pg2, L10 – step or steep increase? Step – it was a rapid event, hence step increase.
Pg2, L21 – Additional references:
*Goldberg, D. N., Gourmelen, N., Kimura, S., Millan, R., & Snow, K. (2019). How Accurately Should We Model Ice Shelf Melt Rates? Geophys. Res. Lett., 46 (1), 189{199. doi: 10.1029/2018GL080383*
*Pattyn, F., Favier, L., Sun, S., & Durand, G. (2017). Progress in Numerical Modeling of Antarctic Ice-Sheet Dynamics. Current Climate Change Reports, 3 (3), 174{184. doi: 10.1007/s40641-017-0069-7*
Done
Pg2, L31 – Rephrase sentence starting "The geometry of LCIS…". I had to read it twice as it sounds like the specific locations were measured by inverting gravity data, rather than the gravity inversions being used to help choose targets for the specific measurements. Done
Pg3, L5 – Reference to Figure 1 (blue dots)  Done
Pg3, L6 – Add Reference to Nicholls et al., 2012 when boreholes are mentioned. Done
Pg3, L11 – Consider changing title to "New Data Acquisition for clarity. Done
Pg3, L12 – Reference to Figure 1 (yellow dots?) – see general comments. Done
Pg3, L14 – Does digging the plate in "improve source consistency" or coupling? It sounds odd to use "consistency" here, as you have described two different methods of placing the source.
We have clarified this point. We improve consistency between shots at that location for stacking.
Pg4, L15 – Explain the 30 m offset is between the source and the first geophone. Consider moving this a few sentences later, where you introduce the source, rather than here where you are talking about receivers. We have clarified this.
Pg3, L18 – "using a geophone trigger adjacent to the hammer plate" add "to start the recording" or something similar. Done
"A stack on 10 hammer blow were also…" – I'm not quite clear on what this means? Were 10 of the 20 hammer blows stacked to evaluate reflection strength, or were an additional 10 blows made and stacked on site for evaluation? A little re-phasing needed here, as the sentence seems a bit lost.

Done. The 10-blow stack is additional, to allow on-site evaluation of signal quality. Ice base reflections may not be immediately obvious in single-shot data.

Pg3, L22 – See general comments above, I am not clear on where the "supplementary surveys" are on Figure 1. Done

Pg3, L30 – "constrain arrivals" – I think "identify" would be a better word to use here, as you state that travel times were measured on the raw gathers so semblance and AGC wasn't actually used to constrain them? Good point. Done, thanks.

Pg4, L1 – Nice idea to use the multiples in these cases! Thanks!

Pg4, L10 – Are these the "BAS refraction sites" in Figure 1 or all refraction measurements? If so, reference here. As above, some confusions with which data set is which. This has been clarified in the text and referenced to Table 1.

Pg4, L26 – Add reference to Table 2, after "At site PRHB4" Done

Pg5, L29 – "We interpolated all available" change to "We gridded all available" or "We interpolated between all available" Done

Pg6, L6 – Errors on the gridded product is potentially much larger that the errors quoted in Section 3.3 - a comment to that effect here would be good. We have added a comment to this effect.

Figure 1: As mentioned above. I am confused with the labelling of survey data here, compared to the text in Section 3.1 and Table 1: Are blue points those from Brisbourne et al., 2014? Are the yellow points a combination of MIDAS, SOLIS and RACE and new reflection surveys? What are the red points, just BAS refraction or ALL refraction surveys? Please clarify. Figure 1 has been modified to allow discrimination of the surveys with additional references in the body of the manuscript.

Figure 2: Add labels (e.g. P1, M1, P2, M2) next to diagrams on right hand side to signify which are multiples and which primaries – something similar to Brisbourne et al., (2014). It might not be clear to those without a seismic background what they are are looking at. Done

Table 1: Add column for reference to paper where data is presented, where relevant. This is seen as unnecessary as the only data which were published previously are the BAS refraction data (Brisbourne, 2014) but which are included here as they form part of the new analysis.

**Anonymous reviewer #2**

**General comments**

The authors present new point seismic data collected beneath the Larsen C Ice Shelf in West Antarctica. These new data are used to produce a bathymetric grid of the seabed beneath the ice shelf and some of the main implications of features revealed in this new grid are discussed. This data set is crucial to improving modelling efforts in an important and rapidly changing region, however I find the paper to be very light on important details and I have some concerns about the bathymetric grid that is presented. These major issues are listed below and are followed by more minor technical corrections.

**Specific comments**

• The paper is very light on details and this is a particular weakness in terms of the error estimates and gridding as detailed below, but also throughout the paper in general.

We acknowledge that Reviewers 2 and 3 found the manuscript to be light on detail and recognise that this was due to building on previous comprehensive studies with more exhaustive data sets. To this end, we have added more detail to make this paper a more robust standalone work. Details below.

For example, I think a section discussing the problems with other estimates of sub-shelf cavity and the difficulties in obtaining these measurements is worthwhile.

We have added a paragraph at the end of the introduction to discuss the alternative methods.

Section 5 is very brief and could greatly benefit from more careful analysis and discussion, particularly in the context of future ice-shelf stability. Furthermore I think there should be more care to emphasise the weaknesses of the grid and the interpretation that follows from it in areas where only one or two data points are available.

We regard detailed discussions of the implications of the results on future ice shelf stability as speculation without a significant amount of work to model ocean circulation beneath the ice shelf and therefore do not go into any more detail.

Finally, many parts of the paper are completely missing references.

We have updated the paper with a number of additional references.

• Section 3.3 seems to rely entirely on analysis made in previous studies and makes no attempt that I can see to constrain uncertainties using the newly collected data.

Indeed, uncertainties are discussed in detail in Brisbourne at el. (2014) upon which this paper builds. We have augmented discussions here with individual uncertainties for each new measurement and included these in Table 2.

How are the picking errors estimated?

We have added this to the text and included a lot more detail on uncertainties (S3.3).

The assumption of a linear variation in ice temperature from 100m below the surface to the base of the ice shelf is almost certainly flawed when you consider a typical ice shelf temperature profile.

A linear temperature profile is a reasonable fit to the measured temperature profile in the ice column (ice column temperature data acquired by Nicholls et al. 2012 - unpublished). In addition, at this northerly latitude the temperature range, and therefore velocity range, is low and therefore this assumption is insignificant. Text has been added to the manuscript to explain this (P4L33).

Why is a comparison not made between ice thicknesses obtained through surface and seismic measurements, surely this is easily done and worthwhile since in some cases the former is used rather than the latter.

A comparison of ice thickness determined using elevation and seismic methods was carried out previously and presented in Brisbourne et al. (2014). Due to the smaller number of new data points presented here we do not repeat this analysis, which has already proven this approach.

How is the GPS error determined?

As stated in the original manuscript this is the measurement error calculated from the raw GPS data (P6L5).

• My main concern is with the bathymetric grid itself. The paper in its current form presents this grid, rather than individual point measurements, as the main result. Section 4 that describes the gridding does not go into sufficient detail. What grounded-ice-depth measurements are added into the grid, those from Bedmap2? So the gridded grounding line position is consistent with that dataset? The authors state that the ice draft in Bedmap2 is better constrained, do they mean the draft as calculated from floatation in regions far away from their own measurements? If so the last sentence is wrong since these thickness values in Bedmap2 are not measurements and therefore not 'known'.

We regard the seismic measurements as the fundamental product which is being presented here and these form the bulk of the data in the repository. The gridded product is included to put these data into context and highlight their value as without this the data would have little meaning or value to the reader. We have added detail of which Bedmap2 data are used (essentially outside of the cavity) and the gridding algorithm. It is correct to state that ice thickness is not "known" but comes from altimetry and floatation calculations. This has been corrected. (S4).

I find the choice of natural neighbour interpolation very puzzling and I think this choice will dramatically affect the resulting grid. I don't think the authors' justification is sufficient, but if there is a very good reason for this choice I would like a discussion and appropriate references. Many deep bathymetric points near the grounding line end up as bullseyes in the grid whereas they are almost certainly located in the troughs of paleo-ice-streams.

This argument arises repeatedly in internal discussions regarding gridding algorithms. Indeed, one method to grid the data would be to use palaeo ice stream routes to delineate troughs. However, by employing such methods one is prescribing geometry using assumptions which may not be valid. We can see how this may be a preferred method for some practitioners and anticipate that some users of

the data may prefer this method. However, our preferred method uses only the data available and does not prescribe geometry using theories or models. In our opinion the natural neighbour method remains true to the observations, where they are available and we regard this as the most appropriate method for the presentation of these data.

Deepening the seabed to ensure an open cavity in some cases is clearly a necessary evil but how extensively is this done? If this is going to be used by ice sheet modellers then I would strongly suggest a much greater minimum thickness than 10m, since even small transient dhdt values during the start of a simulation will cause the ice shelf to ground and completely change ice flow in the region. An uncertainty estimate would be useful but presumably would be difficult to produce without using a more advanced gridding algorithm (which I strongly feel should be done, or at least a comparison made to justify the choice made here).

We agree. Quantifying the uncertainty in a grid is extremely difficult and we have tried with this and other data sets without success. We also agree that using a minimum cavity thickness is a necessary evil and appreciate that the reviewer recognises this. As we do not intend, nor expect, the gridded product to be used directly by the oceanographic community, we used 10 m as this most closely resembles the original interpolated grid in form. The specific minimum depth used is determined by the type and resolution of oceanographic model used and there is therefore no correct value. We now do however include an additional figure (Figure 4) to highlight where this update has been done and to what degree. This is a useful addition and helps to highlight weaknesses in the interpolation and data coverage.

Also given that many of these data are already published elsewhere and lead to the generation of a previous grid, I think a direct comparison is essential to be able to ascertain which features are genuinely new discoveries resulting from the data presented here.

Overall, more detailed discussion is needed. I realise a perfect grid is impossible with limited data, but I do not feel that sufficient care has been taken with grid generation and I think the resulting discussion skips over these issues, particularly given that it is presented as the main result of the paper.

We do not think it necessary to focus on highlighting new features. As mentioned above, data points in previously unmeasured areas which confirm previous interpolations are just as valid as data points which contradict previous results as without confirmation, neither model can be deemed 100% reliable. We therefore discuss the main features and simply highlight whether these are confirmatory or not. Without oceanographic modelling, the importance of individual features cannot be ascertained, and therefore lies beyond the scope of this paper.

**Technical corrections**

Throughout the paper: Hyphenation rules inconsistently applied

We've tried hard to ensure consistency. A consequence of 14 authors I suspect.

p. 1 l. 21: Capitalise West (also in title)

LCIS is in west Antarctica but it is not in West Antarctica, as in the West Antarctic Ice Sheet. It is the Antarctic Peninsula Ice Sheet, hence west with the lower case.

p. 1 l. 28-30 Repeat of earlier parts of the abstract

This sentence specifically describes the data set, the earlier part of the abstract describes the study. We have changed the paragraph structure to reflect this.

p.2 l. 3: Missing references for buttressing e.g. Rott 2002, Furst 2016, Reese 2018

Rott and Furst added but we don't see Reese (2018) as appropriate.

p.2 l. 14: Missing references for Antarctic Peninsula warming

Good point. Vaughan et al. (2003) added.

p. 2l. 26: accurately predict This just depends on whether we split the infinitive.

p. 3 l. 9: here and elsewhere should be bathymetric when used as adjective and bathymetry when a noun. Good point but not necessarily always the case. In some cases we use the compound noun, such as "bathymetry map" where this is a specific entity.

p. 3 l. 16 Ensure that units are in-line with numbers We will make sure this is resolved at the type-setting stage.

p. 5 l. 1: How were picking errors determined, was a repeat of each pick done?
Correct, consistent with previous studies. Text has been added to clarify.

p. 5 l. 22: What is the vertical GPS accuracy based on, why were some surface elevation measurements not available?
As stated, this is the measurement uncertainty (calculated from the raw data). Not all field campaigns used dual frequency GPS to measure surface elevation. This is one of the complexities of integrating data from different field campaigns made by different groups and Table 1 presents this fact.

p.5 l. 26: Based on all of the above uncertainty sources, we arrive at a cumulative overall uncertainty…
Done

p. 6 ;. 12: Surely another crucial aspect of the bathymetry around this ice rise is whether slight thickening could lead to extensive re-grounding.
Correct, although thickening in this area is not seen as particularly likely in the near future, words have been added to this effect.

p. 6 l. 16-17: I don't see this path of Jason trough north of Bawden Ice Rise since the bathymetry there is shallower than the route south.
Whether this is actually a route for oceanographic circulation is beyond the scope of this paper and an aspect to be addressed by the oceanographic community. We are merely stating that we can confirm that no barrier exists to this deeper water previously modelled in say Nicholls et al. (2012).

p.6 l. 17-19: This is based on one data point and the gridding here has just created a presumably unrealistic bullseye around that point. Presumably the reality is that there is a trough here leading towards the grounding line that is missed in the grid and this should be discussed.
This is of course the ultimate limitation of sparse data coverage and the reason why we present the grid with data points overlain to highlight coverage. Any interpretation of oceanographic models using these data will need to be fully aware of these limitations and discuss them in context. We have added a sentence at the start of this section to highlight this fact.

p. 6 l. 21: References needed.
Done

Fig. 3: Given that this is the main figure of the paper I think it needs a lot more work. Firstly the resolution is far too low for publication. The background MODIS imagery is either completely missing or not discernible. Please highlight the grounding line by making it a bolder contour. What do the black arrows indicate? Add label for Mobiloil inlet and Cole Peninsula which are discussed in the discussion text.

Some good points here. We have updated and improved this image and added additional annotation for clarity as well as the Bedmap2 grounding line. Unfortunately, the resolution is limited by the 5MB limit imposed by the publisher for images. The MODIS imagery was used to highlight the new ice front only and so we have replaced this with a solid line to improve the clarity of the rest of the figure.

**Anonymous Referee #3**
This study presents new bathymetric information derived from seismic shots beneath the Larsen-C Ice cover. The Significance of new information has no doubt concerning ocean circulation and global climatological issues. However, it is not very clear to me up to which level your new data improve previous datasets. The main global comment that I may have from reading the paper several time is that further care should be taken on making the difference on the contribution of the existing data and the motivation/input from the new data.
As stated in the general response, even if new data do not delineate new features, the very fact that they can be used to confirm the existence of previously "assumed" features is a valuable result when modelling sub-shelf bathymetry. We are therefore not particularly concerned about highlighting lots of new features which may only be marginally different from the previously assumed features.

Specific comments –
Location of previous work: I believe that a table giving summary statistics of the previous dataset would be valuable (columns could be like: survey name, survey date, type/sensor, number of measurements, estimated vertical accuracy, estimated horizontal accuracy. - Using the table proposed above you can detail the new dataset –

The data we feel are required are already presented in Table 1 (P13). More detail is provided in the metadata of each archived data set.

It is not very clear to me how the gridding methodology is done. I understand you've used natural neighbour interpolation. You should provide a schema of your procedure in which we could see the data flow, the different steps (data preparation, gridding process, corrections) and the parameters used. –

Details and discussion added in Section 4.

Your gridding correction steps (line 2-5, p6) is not clear. It looks to me as some sort of data tweaking. Please see reviewers #2 on this point. –

As reviewer #2 accepts this "tweak" is a necessary evil of transforming the grid where data are sparse a form that can be used by the oceanographic community. We have outlined our procedure to ensure this is understood by end users of the gridded product. We have added an additional figure (Figure 4) to highlight areas where the mismatch between the interpolated grid and ice draft is most prevalent, mostly along the grounding line.

Concerning the two last points I believe that the minimal aim of your paper and more specifically section 4 is to enable any readers/data user to be able to reconstruct the bathymetric grid. Therefore I suggest being more explicit in your gridding methodology. Algorithm, implementation, software, parameters

As described above, the main aim of our paper is to present the data and discuss its potential significance. The gridded product is not seen as an essential output for the community. We have however added enough detail to allow the reader to reproduce the gridded product but also highlighted that we do not see it as a final product.

I do not pretend to be able to comment on the English or the style; however I would suggest limiting the vagueness to its minimum. You should be more explicit and limit yourself from using terms like "relatively", "more reliable", "where required", "consistent", "much lower", "further uncertainty",

We have removed a number of these issues where we feel they are not essential. We would argue that the term "relative" is in certain instances valid, for example.

**Reviewer #4 Coen Hofstede (Referee)**
General comments: The paper presents addition of seismic point measurements to existing ice column–water column thickness measurements at Larsen C Ice Shelf (LCIS). By measuring the travel times of the ice shelf base and seabed, the ice shelf thickness and water column are calculated. In addition to existing data points and bedmap2, a sub-shelf bathymetric map of LCIS is created. The paper is well written, clearly built-up, and easy to follow. The method to calculate ice thickness and water column is straightforward and well explained, but certain parts are described vaguely, making it hard to judge the quality of the presented data, such as the p-wave velocity of the ice column and the gridding process.

The bathymetric sub-shelf map of LCIS is a valuable addition to the gap in bathymetric data under Antarctic ice shelves. The data points better constrain the sub-shelf bathymetry such as their key findings show, and improve the modeling of the oceanshelf interaction. LCIS is probably next in line to disintegrate.

**Specific Comments:**

3.2/3.3 Seismic velocities/Uncertainties The velocity analysis of the firn/ice column is well explained but a number (or range) for the ice velocity(ies) would be nice. Indirectly this is mentioned at the uncertainty of the ice column thickness, being 3.8m at 1 ms uncertainty, which suggests the ice velocity is 3800m/s. If that is the case I come to half of the suggested uncertainty as the times of reflections are TWTs. To understand the uncertainty of the ice column, velocities of the ice column are essential.

The range of maximum velocities is now included in Section 3.2. Uncertainties are now presented and discussed in more detail and individual uncertainties presented in Table 2 for each new seabed depth measurement.

It is not clear if the measurements are corrected for tides, I suspect not. This is important for those shots that do not show no ice base return. With a tidal range of 2 m, I would come to approximately 20m inaccuracy. How many shots do not have this ice base return, one at PRHB4 or more?

As already outlined at P5-L23 in the original submission, no correction for tides is made, consistent with Brisbourne et al. (2014). This is included in the uncertainty estimate. It is not clear to us how a 2 m tide will result in 20 m inaccuracy. Indeed, as stated in the original manuscript, only PRHB04 lacks an ice base reflection in the seismic data.

Although the error analysis is clearly described and the order of magnitude is correct, the choice of 10m accuracy seems somewhat arbitrary to me. Why not 9m or 13m I wonder?
We now provide uncertainties calculated for each site separately (Table 2).

Bathymetric gridding I think it is important here to be clear about the gridding method is used rather then "which is well suited to a dataset with an uneven distribution of data point". It is important to know how you get from data points to the gridded bathymetry map. A reference possibly?
Additional text has been added to Section 4 to provide this detail.

I find the phrasing about the gridding problem at places where the "calculated seabed is shallower than : : :..the ice draft of the Bedmap2 dataset" unclear. Are these calculations ignored or overruled by a deeper seabed? If so it would make sense to mark these data points in map 3 so that we know exactly what data points have been used in the gridding
We have replaced the word "calculated" with "interpolated" to help clarify this issue. The problem arises where the interpolated values are contradicted by the measured ice thickness data, not where we have direct measurements of sea bed depth.

Figure 1: The text (3.1) and Table 2 mention 30 measurements (14 seismic bathymetry measurements and 16 seismic refraction and reflection surveys). In the figure I see 28 yellow dots (new measurements) and 3 red dots. - How do these 28+3=31 dots relate to the 30 measurements from Table 2? Please explain in the caption or adjust the figure.
Figure 1 has been updated to make this clearer. Two MIDAS sites are collocated and one was mis-labelled as it is a repeat of an existing site.

Figure 3: Please use another color for the contour lines. They can hardly be made out.
Done

**Technical corrections:**

Table 1, receiver spacing MIDAS: Why an asterisk? Well spotted! These are the data referred to in the caption where the acquisition geometry varies and as such is detailed in the data repository.

---

## Author Response (AR2)

**An updated seabed bathymetry beneath Larsen C Ice Shelf, west Antarctica by Alex Brisbourne et al.**

**Response to Editor - March 2020**

We have made three minor edits (two were requested by the editor, the third we feel will make the manuscript more likely to be found by the relevant the community).

1 – P1L1 - Title changed to "Antarctic Peninsula" from "west Antarctica"

[revised manuscript text omitted]